# Self-Assessment of Oral Health-Related Quality of Life in People with Ectodermal Dysplasia in Germany

**DOI:** 10.3390/ijerph16111933

**Published:** 2019-05-31

**Authors:** Marcel Hanisch, Sonja Sielker, Susanne Jung, Johannes Kleinheinz, Lauren Bohner

**Affiliations:** Department of Cranio-Maxillofacial Surgery, Research Unit Rare Diseases with Orofacial, Manifestations (RDOM), University Hospital Münster, Albert-Schweitzer-Campus 1, Building W 30, D-48149 Münster, Germany; sonja.sielker@ukmuenster.de (S.S.); susanne.jung@ukmuenster.de (S.J.); johannes.kleinheinz@ukmuenster.de (J.K.); lauren@usp.br (L.B.)

**Keywords:** rare diseases, oral health related quality of life, OHRQoL, ectodermal dysplasia, OHIP-14, patient-related outcome

## Abstract

*Background:* Ectodermal dysplasia describes a heterogeneous group of hereditary, congenital malformations involving developmental dystrophies of ectodermal structures. The aim of this study was to analyse the oral health-related quality of life (OHRQoL) in people with ectodermal dysplasia and to evaluate the influence of different variables. *Methods:* The study was designed as an anonymous epidemiological survey study among people with ectodermal dysplasia to evaluate oral symptoms, satisfaction with the health system and their respective OHRQoL using the validated German version of the OHIP-14 (Oral Health Impact Profile) questionnaire. *Results:* When asked about oral symptoms, 110 of the participants provided responses, of which 109 (99.09%) described oral symptoms. The average age of the female participants at the time of diagnosis was 17.02 years (range: 0 to 48 years), the average age of men was 5.19 years (range: 0 to 43 years). The average OHIP-14 overall score for female participants was 12.23 points (SD: 12.39), for male participants an average OHIP score of 11.79 points was recorded (SD: 11.08 points). Difficulty in finding a dentist (*p* = 0.001), and the dissatisfaction with the health system (*p* = 0.007) showed a negative impact on the OHRQoL. *Conclusion:* People with ectodermal dysplasia rate their OHRQoL worse than is usually prevalent in the normal German population (4.09 points); women are diagnosed with “ectodermal dysplasia” later than men. Participants who reported difficulties in finding a dentist for treatment exhibited higher OHIP values. Likewise, dissatisfaction with the health system demonstrated a negative impact on the oral health-related quality of life.

## 1. Introduction

Ectodermal dysplasia describes a rare heterogeneous group of hereditary, congenital malformations involving developmental dystrophies of ectodermal structures affecting about one in 5000 to 10,000 people [1].

Ectodermal derivatives include hair, teeth, nails, sweat glands, sebaceous glands, mammary glands and the eyelash glands [2,3,4,5]. In addition to syndromic forms, monosymptomatic forms involving oligodontia are also described [6]. In the meantime, almost 200 clinically or genetically distinct forms of ectodermal dysplasia have been recorded [2,4,7].

Oral manifestations of the various subtypes of ectodermal dysplasia not only include dental agenesis (hypodontia, oligodontia, anodontia), shape anomalies of the teeth such as microdontia or conical teeth [8] or reduced salivary flow rates [9,10], but also subtypes, including cleft formation, like EEC syndrome (ectrodactyly-ectodermal dysplasia-cleft lip/palate syndrome) [11]. Manifestations of ectodermal dysplasia in the tooth, mouth, jaw and facial areas can negatively influence chewing, swallowing and speech functions [12] and lead to psychological impairments, especially in patients with severe oligodontia [13].

Our research group showed previously that the oral health-related quality of life (OHRQoL) is affected by rare diseases presenting oral manifestations [14,15,16,17].

However, so far, there is hardly any data on OHRQoL in people with ectodermal dysplasia. Therefore, the aim of this study was to obtain information on OHRQoL in people with ectodermal dysplasia, their satisfaction with dental care in the health care system and the diagnosis period between the first symptoms of the disease and their diagnosis.

## 2. Methods

### 2.1. Study Design

The study design used in this study has already been described in previous works [15,16,17]. The study was designed as a cross-sectional questionnaire cohort study in people with ectodermal dysplasia to evaluate their respective OHRQoL. A questionnaire was developed for this purpose, which consisted of questions on age, sex, age at diagnosis, the time period between the first symptoms and the diagnosis of the disease, oral symptoms, difficulties in finding a dentist for treatment and satisfaction with the level of support provided by the German health care system with regard to dental care. Furthermore, the validated German version of the OHIP-14 (Oral Health Impact Profile) questionnaire was used to evaluate each individual’s OHRQoL [18].

In summary, each of the 14 questions of the OHIP-14 questionnaire was assigned a standardised numerical value: 0 = never, 1 = rarely, 2 = from time to time, 3 = often and 4 = very often. The highest score was 56 points and indicated the worst OHRQoL. The questions referred to experiences during the past month [18].

The questionnaire was presented at the annual meeting of the German support group “Ektodermale Dysplasie e.V.” on 28th April 2018 in Rehe, Germany. The questionnaire was subsequently sent to members by the support group as a supplement to the members’ magazine in May 2018, together with an information sheet and the conditions for participation in the study. Generally, the magazine informs about activities of the self-help group and activities in the field of ectodermal dysplasia (e.g., politics, medicine, health system). The aim was to collect at least 110 completed questionnaires, which corresponds to half of the potential participants. The desired number of responses was reached on 15th November 2018.

The present research has been approved by the ethics committee of the Ärztekammer Westfalen-Lippe and the Westfälische Wilhelms Universität Münster (Ref. No. 2016-006-f-S).

### 2.2. Participants

Adult participants aged 16 and above were included in this study. The self-help group “Ektodermale Dysplasie e.V.” specified the number of members affected who have reached at least 16 years of age as *n* = 220.

### 2.3. Statistical Analysis

Statistical analysis was performed using the SPSS 22.0 software (IBM Corp., Armonk, NY, USA). Descriptive statistics were presented as mean (standard deviation), median (interquartile range) and 95% confidence interval (95% CI). The OHIP scores were calculated for different groups: gender, difficulties in finding a dentist, satisfaction with the health system, participants age, age at diagnosis, time between first symptom and diagnosis. The Kolmogorov-Smirnov test was used to evaluate the normality of the data. Mann-Whitney and Kruskal-Wallis tests were used to compare the difference among groups. Additionally, the Spearman test was used to evaluate the correlation between the OHIP values and quantitative data.

## 3. Results

A total of 110 questionnaires were evaluated, consisting of 40 female (36.36%) and 70 male (63.63%) participants. According to the support group “Ektodermale Dysplasie e.V.”, 220 members fulfilled the inclusion criteria, resulting in a response rate of 50%, which was initially defined as the target. The gender-independent age average was 33.65 years (range: 16–67). The average age of the female participants was 36.57 years (range: 16–67). The average age of the male participants was 31.95 years (range: 16–64).

The gender-independent average age at which ectodermal dysplasia was diagnosed was 9.58 years (range: 0 to 48 years). The average age of the female participants at the time of diagnosis was 17.02 years (range: 0 to 48 years), the average age of men at the time of diagnosis was 5.19 years (range: 0 to 43 years). The period between the first signs of their disease and their diagnosis was 9.04 years (range: 0 to 48 years), the average age of the female participants was 15.55 years (range: 0 to 48 years), for men 4.96 years (range: 0 to 43.5 years).

When asked about oral symptoms, 110 of the participants provided responses, of which 109 (99.09%) described oral symptoms. With respect to oral symptoms, 108 participants named dental agenesis (hypodontia, oligodontia, anodontia). Shape anomalies of the teeth, such as peg-shaped teeth or microdontia, were mentioned by 59 of the respondents (53.63%). Different types of dysgnathia (micrognathia, progenia, microretrognathia) were mentioned by 12 participants (10.90%). Mineralisation disorders of the tooth structure were reported in 10 cases (9.09%), and reduced salivary flow rates, such as xerostomia and hyposalivation, were reported four times (3.63%). Orofacial clefts were reported by eight participants (7.27%) (Figure 1).

In our survey, 103 of the respondents provided information on their satisfaction with the support they received (services offered and financed by the health system) with dental care provided by the German health care system. Of these, 76 study participants were dissatisfied with the health care system (73.78%), and 26.22% (*n* = 27) were satisfied.

Of the 106 participants who provided data on this topic, a total of 29 participants had difficulties to find a dentist to treat them (27.35%), and 72.65% (*n* = 77) had no difficulties.

The average OHIP-14 overall score for female participants was 12.23 points (SD: 12.39). For male participants, an average OHIP score of 11.79 points was recorded (SD: 11.08). In the group that was satisfied with the health care system, the average OHIP score was 6.92 points (SD: 7.48), among participants who were not satisfied with the health care system, an average score of 13.91 was reported (SD: 12.22). The average OHIP scores (standard error) for gender and satisfaction with the health system are shown in Figure 2 and Figure 3.

A separate analysis of the three individual dimensions of the OHIP-14 showed a mean value for both genders in dimension 1 (functional limitations) at 3.27 points, in dimension 2 (psychological aspects) at 6.41 points and in dimension 3 (physical aspects) at 2.27 points.

Participants who reported difficulties to find a dentist, as well as those unsatisfied with the health system, showed higher OHIP values, which represents a negative impact on the oral health-related quality of life. The participants’ gender did not influence the OHIP values (Table 1). There was no correlation between OHIP values and the participants’ age (0.063; *p* = 0.527), age at diagnosis (−0.061; *p* = 0.533), or the time between first symptom and diagnosis (0.105; *p* = 0.315).

## 4. Discussion

A total of 110 participants allows for conclusions to be drawn about the OHRQoL in patients with ectodermal dysplasia in Germany. Almost all participants reported dental agenesis, and more than half of the participants reported shape anomalies of the teeth, which have already been documented in the respective literature [6,8,19].

It could be demonstrated that both female and male participants are affected by increased OHIP scores (mean female: 12.23; mean male: 11.79). Comparative data must be used to classify the values. In an epidemiological study, also using the validated German version of the OHIP-14 questionnaire, an average OHIP-14 score of 4.09 points was established for the average German population [20]. Consequently, it is evident that people with ectodermal dysplasia have a significantly worse OHRQoL than the average population.

On this basis, it can be deduced that health policy demands additional support for patients with genetically caused dental agenesis of the type present in ectodermal dysplasia. In some health care systems, such as the German health care system, it is possible to provide patients with implant-mediated prosthetics at the expense of the public social systems via exceptional statutory provisions, e.g., if the patients are psychologically affected by ectodermal dysplasia [21]. Expert centres should also be established for the complex implant-prosthetic restoration of patients with oligodontia. The costs of the care should be largely financed by the health care system.

The dentist is therefore not only responsible for identifying genetically caused dental agenesis and shape anomalies [8], classifying them and, if necessary, initiating further diagnostics (e.g., human genetic clarification), but also for drawing the attention of affected persons to prosthetic care options, potentially even at the expense of the public health system.

The resulting demands on the health care system and dentists are further confirmed by statistical analysis of the available data. Thus, participants who reported difficulties in finding a dentist showed statistically significant higher OHIP values (*p* = 0.001). Likewise, the dissatisfaction with the health system showed a statistically significant negative impact on the OHRQoL (*p* = 0.007).

Dentists, oral/maxillofacial surgeons and orthodontists seem to play a key role in recognising patients with genetically caused dental agenesis and other anomalies, such as ectodermal dysplasia. Often, especially the orthodontist is likely to be the first (dental) physician to be contacted wherever dental agenesis arises [22]. Consequently, the orthodontist plays a decisive role in the recognition of patients with ectodermal dysplasia which do not exhibit any external symptoms at first glance (Figure 4, Figure 5, Figure 6, Figure 7, Figure 8 and Figure 9).

Fortunately, most of the participants (72.65%) reported that they had no difficulty finding a dentist to treat them. We could already prove in another study that people with rare diseases in Germany had no problem to find a dentist [16], even if the values now determined here were somewhat worse than in the investigation at that time (79.47%). This may be due to the complex dental therapy that some patients with oligodontia require (e.g., bone augmentation, implants, prosthesis). Perhaps, participants who reported difficulties in finding a dentist therefore reported worse OHIP values.

It was especially noticeable that male participants reported being diagnosed earlier than female participants. One explanation could be that mostly female patients are affected by externally inconspicuous subtypes (Figure 4, Figure 5, Figure 6, Figure 7, Figure 8 and Figure 9) and are therefore diagnosed later. However, an additional study would be necessary to evaluate this clinically.

### Study limitations

The results of this study are based on the data provided by the study participants. The study and the conditions for participation were presented at the annual meeting of the support group and presented again in an information letter attached to the questionnaire. However, participants were not clinically examined. A further study which would clinically examine the participants would be helpful.

## 5. Conclusions

People with ectodermal dysplasia rate their OHRQoL worse than is usually prevalent in the normal German population; women are diagnosed with “ectodermal dysplasia” later than men. Participants who reported difficulties in finding a dentist for treatment exhibited higher OHIP values. Likewise, dissatisfaction with the health system demonstrated a negative impact on the oral health-related quality of life.

## Figures and Tables

**Figure 1 ijerph-16-01933-f001:**
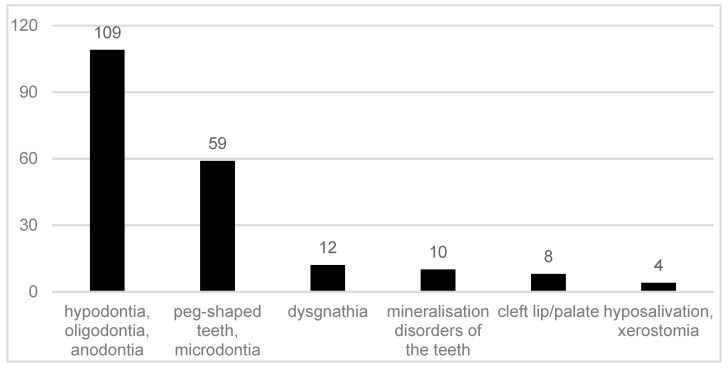
Reported oral symptoms in 109 participants with ectodermal dysplasia.

**Figure 2 ijerph-16-01933-f002:**
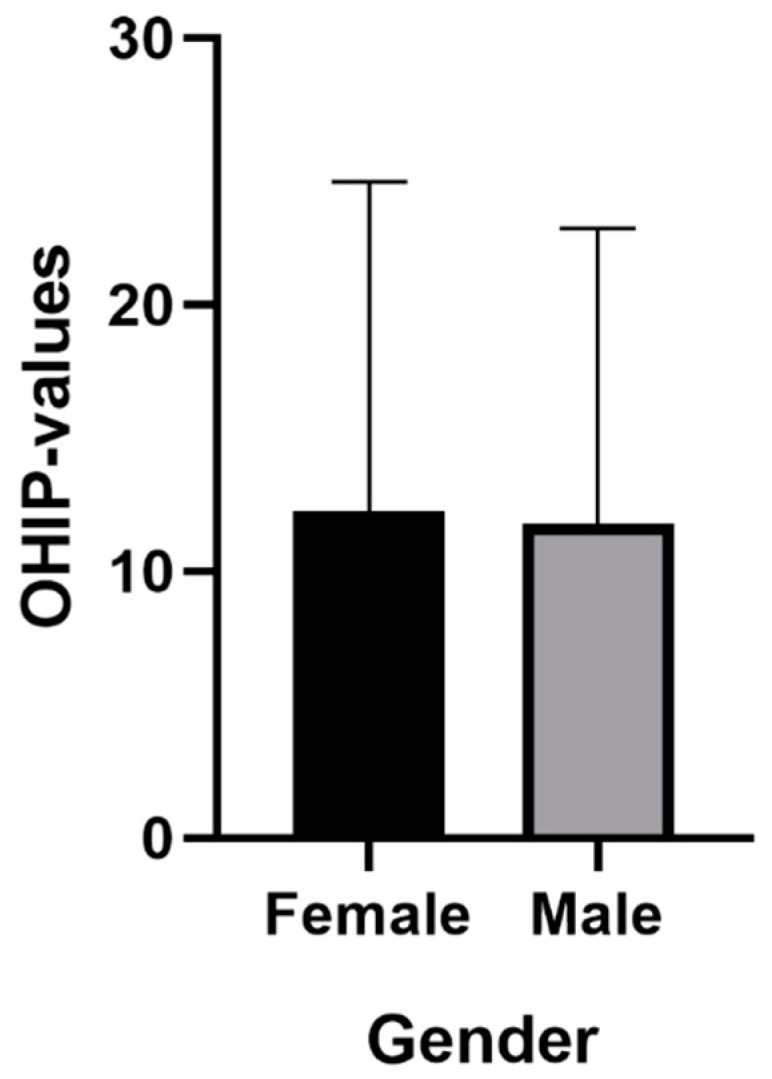
Mean (standard error) for gender.

**Figure 3 ijerph-16-01933-f003:**
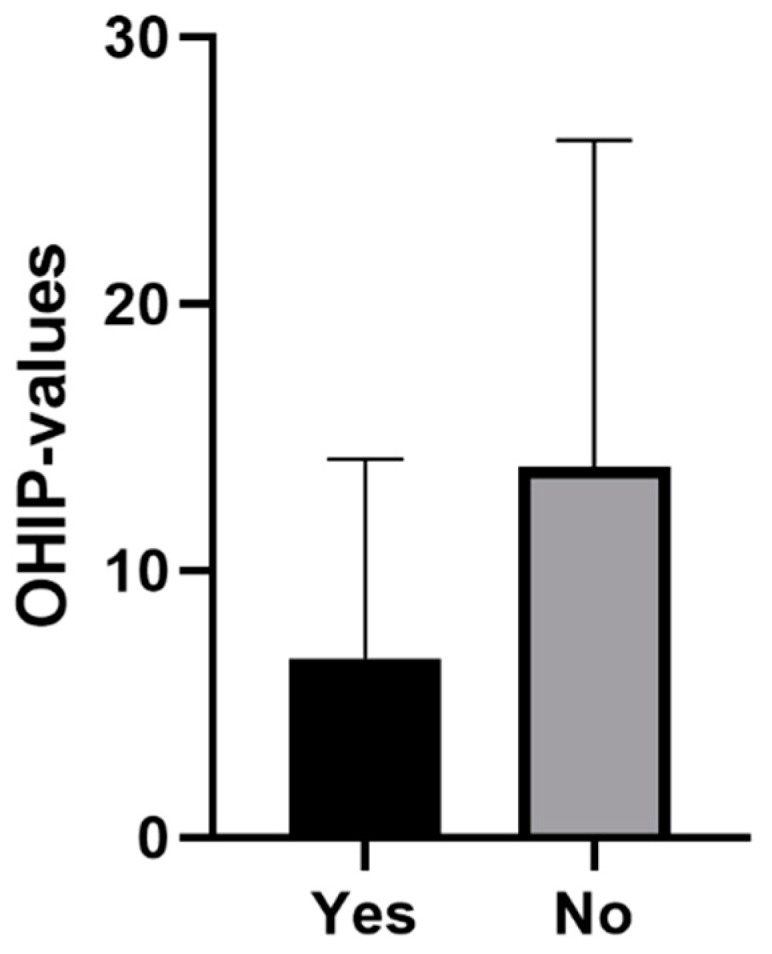
Mean (standard error) for satisfaction with the health system.

**Figure 4 ijerph-16-01933-f004:**
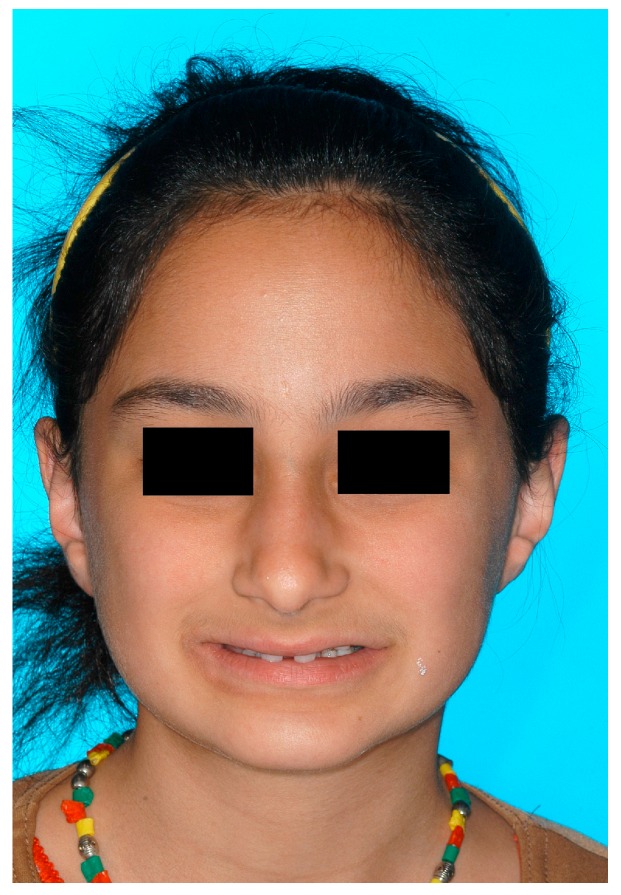
A 12-year-old female patient, no visible signs of ectodermal dysplasia, with ectodermal dysplasia.

**Figure 5 ijerph-16-01933-f005:**
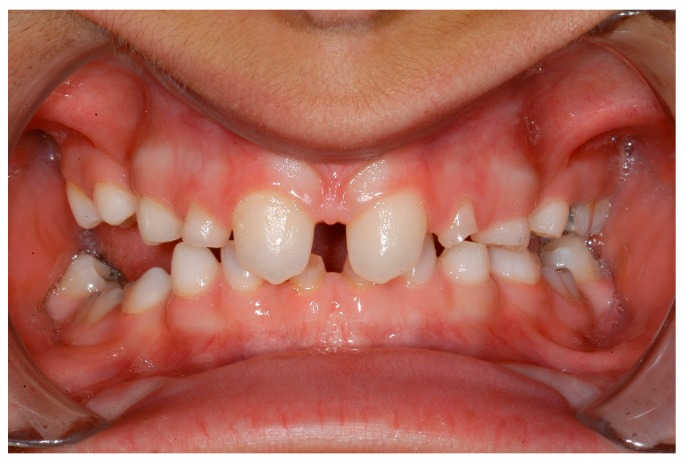
Intraoral situation: oligodontia, microdontia of teeth 11 and 21 and a pronounced lateral open bite.

**Figure 6 ijerph-16-01933-f006:**
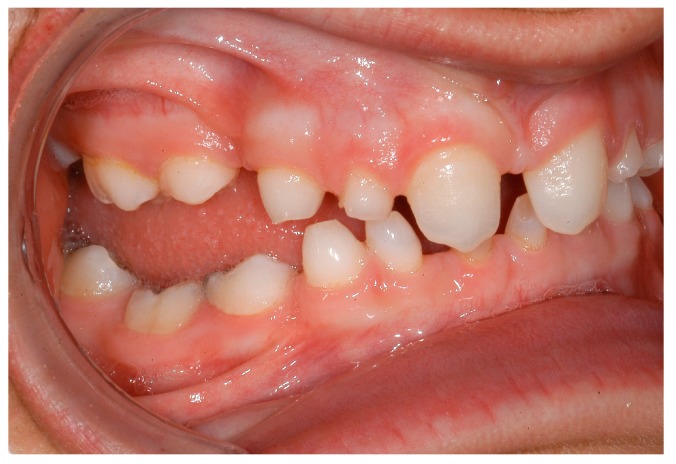
Intraoral situation: oligodontia, microdontia of teeth 11 and 21 and a pronounced lateral open bite.

**Figure 7 ijerph-16-01933-f007:**
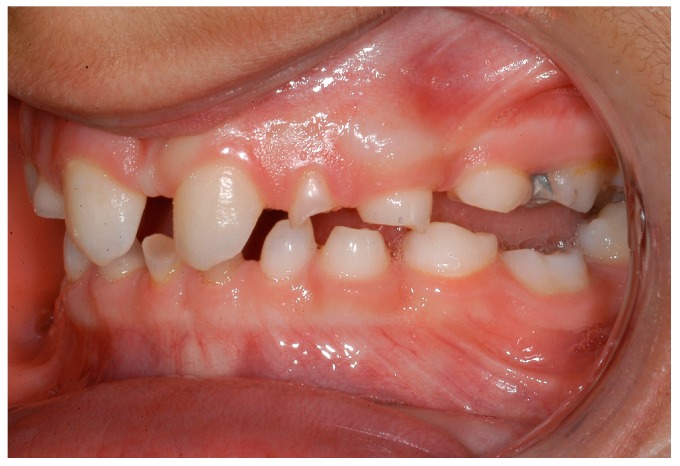
Intraoral situation: oligodontia, microdontia of teeth 11 and 21 and a pronounced lateral open bite.

**Figure 8 ijerph-16-01933-f008:**
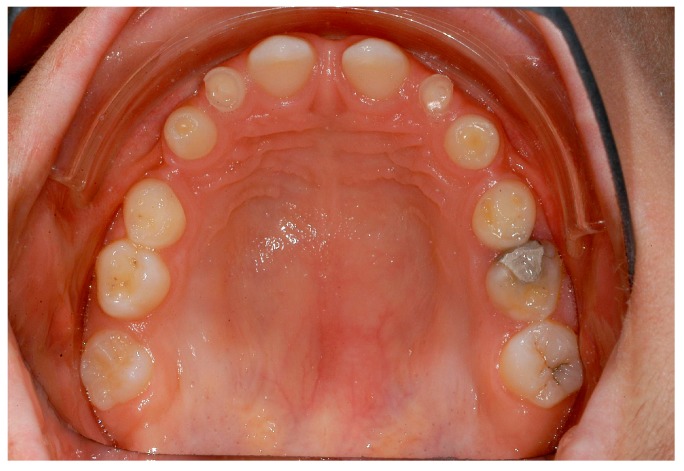
Intraoral situation: oligodontia, microdontia of teeth 11 and 21.

**Figure 9 ijerph-16-01933-f009:**
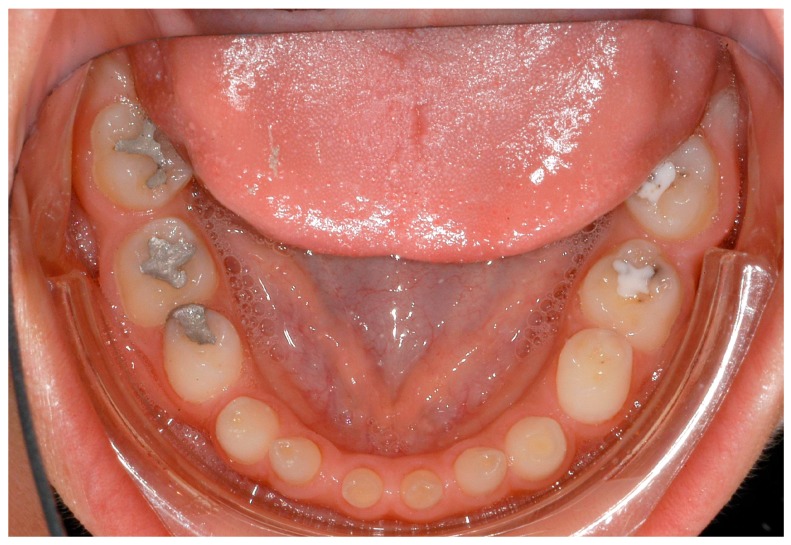
Intraoral situation: oligodontia, microdontia of teeth 11 and 21.

**Table 1 ijerph-16-01933-t001:** Descriptive statistics from OHIP (Oral Health Impact Profile) values.

	Mean (Standard Deviation)	Median (Interquartil Range)	95% CI	*p*-Value
Gender				0.781
M	11.79 (11.08)	9 (17.25)	(8.97; 14.60)	
F	12.23 (12.39)	8 (18)	(7.91; 16.55)	
Difficulties in finding a dentist				0.001 *
No	9.21 (8.58)	5 (13)	(6.94; 11.47)	
Yes	19.72 (13.06)	19 (20.50)	(14.32; 25.11)	
Satisfaction with the Health System				0.007 *
No	13.91 (12.22)	10 (20)	(10.97; 16.84)	
Yes	6.92 (7.48)	5 (11)	(3.96; 9.88)	

* means statistically significant.

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
