# Peer review of "Self-Assessment of Oral Health-Related Quality of Life in People with Ectodermal Dysplasia in Germany"

_ijerph, 2019, doi:10.3390/ijerph16111933_

Round 1
Reviewer 1 Report
This is a really nice epidemiological survey study evaluating Oral health related quality of life among people with ectodermal dysplasia with a validated questionnaire. Data of 110 affected individuals were included representing 50% of affected in Germany, which is one of the strengths of this study.
I have some minor suggestions:
Abstract: Could you please give comparative OHIP-14 scores already in the abstract (like the average OHIP-14 score for the German standard population or the range of possible scores).
General comments
From the OHIP-14, there should be much more data available on functional limitations, and psychological and physcial aspects. Please give details on these data. I am really curious on these data!
Discussion
Please discuss, whether “finding a dentist” and „dissatisfaction with the Health System“ is a special problem for individuals with ED or a general problem in Germany.
Please give practical advices how the satisfaction with the Health System can be improved. Do we need special centres where people with EDS should be treated or do we need treatment guidelines for the general dentists how to treat them? Do people with ED get support from the insurance when they need implants? Do they need a genetic diagnosis to get support?
Author Response
We would like to thank the Editor and the Reviewers for revising our manuscript [ijerph-509304] entitled “Self-assessment of oral health related quality of life in people with ectodermal dysplasia“ and the constructive points discussed. The helpful comments and suggestions for improving the manuscript have been incorporated into the revised version and all changes were highlighted “Track Changes”. In this letter, we provide a point-by-point response to each addressed comment and hope the manuscript is now suitable for publication in the International Journal of Environmental Research and Public Health.
Reviewer 1:
This is a really nice epidemiological survey study evaluating Oral health related quality of life among people with ectodermal dysplasia with a validated questionnaire. Data of 110 affected individuals were included representing 50% of affected in Germany, which is one of the strengths of this study.
I have some minor suggestions:
Abstract: Could you please give comparative OHIP-14 scores already in the abstract (like the average OHIP-14 score for the German standard population or the range of possible scores).
We added the average value of the german normal population in the abstract (line 25).
General comments
From the OHIP-14, there should be much more data available on functional limitations, and psychological and physcial aspects. Please give details on these data. I am really curious on these data!
In the present study, only the total score was evaluated, since a separate evaluation of all three dimensions is not recommended [John MT, Miglioretti DL, LeResche L, Koepsell TD, Hujoel P, Micheelis W. German short forms of the Oral Health Impact Profile. Community Dent Oral Epidemiol. 2006 Aug;34(4):277-88]. We have now listed the mean values of all three dimensions in the results section (line: 129-131)
Discussion
Please discuss, whether “finding a dentist” and „dissatisfaction with the Health System“ is a special problem for individuals with ED or a general problem in Germany.
We now discuss this in the discussion section:
„Fortunately, most of the participants (72.65%) reported that they had no difficulty finding a dentist to treat them. We could already prove in another study that people with rare diseases in Germany had no problem to find a dentist [16], even if the values now determined here were somewhat worse than in the investigation at that time (79.47%). This may be due to the complex dental therapy that some patients with oligodontia require (e.g. bone augmentation, implants, prosthesis). Maybe therefore participants who reported difficulties in finding a dentist reported worse OHIP-values.“ (line 211-217)
Please give practical advices how the satisfaction with the Health System can be improved. Do we need special centres where people with EDS should be treated or do we need treatment guidelines for the general dentists how to treat them? Do people with ED get support from the insurance when they need implants? Do they need a genetic diagnosis to get support?
We now discuss this problem in the discussion section:
„In some health care systems, such as the German health care system, it is possible to provide patients with implant-mediated prosthetics at the expense of the public social systems via exceptional statutory provisions, e.g. if the patients are affected by ectodermal dysplasia [21]. Expert centers should also be established for the complex implant-prosthetic restoration of patients with oligodontia. The costs of the care should be largely financed by the health care system.“ (line 178-183)
Reviewer 2 Report
Comments to the authors:
The authors present a study about oral health-related quality of life (OHRQoL) impairment in patients with ectodermal dysplasia (ed). The topic of the paper is important in relation to the study of rare diseases.
Two points should be addressed by the authors.
First, patients with ed often have removable dentures. Thus, OHRQol-impairment in ed patients should be reported and compared in relation to their prosthesis status as it is done in the paper of John et al. (2004): https://www.idz.institute/publikationen/zeitschriftenbeitraege/normwerte-mundgesundheitsbezogener-lebensqualitaet-fuer-kurzversionen-des-oral-health-impact-profile.html.
Second, it would be helpful for the characterization of OHRQoL-impairment in ed patients to report the five most frequently answered questions to know whether ed patients have the same problems as normal patients without ed (just even more) or different problems.
Author Response
We would like to thank the Editor and the Reviewers for revising our manuscript [ijerph-509304] entitled “Self-assessment of oral health related quality of life in people with ectodermal dysplasia“ and the constructive points discussed. The helpful comments and suggestions for improving the manuscript have been incorporated into the revised version and all changes were highlighted “Track Changes”. In this letter, we provide a point-by-point response to each addressed comment and hope the manuscript is now suitable for publication in the International Journal of Environmental Research and Public Health.
Reviewer 2:
Comments to the authors:
The authors present a study about oral health-related quality of life (OHRQoL) impairment in patients with ectodermal dysplasia (ed). The topic of the paper is important in relation to the study of rare diseases.
Two points should be addressed by the authors.
First, patients with ed often have removable dentures. Thus, OHRQol-impairment in ed patients should be reported and compared in relation to their prosthesis status as it is done in the paper of John et al. (2004): https://www.idz.institute/publikationen/zeitschriftenbeitraege/normwerte-mundgesundheitsbezogener-lebensqualitaet-fuer-kurzversionen-des-oral-health-impact-profile.html.
Thank you very much for this important comment. The aim of the study presented here was to conduct a cross-sectional study on OHRQoL in ed. Thus we did not check the status of the dentures in our study, so we cannot answer this question. We think that a clinical examination of the participants would also be useful for this and take this as a suggestion for future studies.
Second, it would be helpful for the characterization of OHRQoL-impairment in ed patients to report the five most frequently answered questions to know whether ed patients have the same problems as normal patients without ed (just even more) or different problems.
In order to better estimate the impairment of OHRQoL in ed, we have now separately listed the results of the three dimensions in the results section (line: 129-131), even if a separat analysis of the dimensions or items is not recommended by the authors of the OHIP-14G [John MT, Miglioretti DL, LeResche L, Koepsell TD, Hujoel P, Micheelis W. German short forms of the Oral Health Impact Profile. Community Dent Oral Epidemiol. 2006 Aug;34(4):277-88
Reviewer 3 Report
This manuscript describes the oral health related quality of life of people with ectodermal dysplasia in Germany. As this journal is international in scope, I would suggest the authors add the location to the title reflecting the focus area.
The study provides with limited information due to the specify cohort selected and the way of collecting data.
Overall, it is written in good English.
Introduction
Lines 43-43: ‘but also forms including cleft formation including EEC syndrome’. What do you mean by ‘forms’? This sentence is confusing. Please revise.
As you mention (lines 47-48) this work has been previously described within several publications of the same research group. Thus, the novelty of the analysis described in this paper is debatable.
Methods
The first paragraph is redundant with the following paragraphs explaining the study design. Please revise and rephrase.
The description of the study design is incorrect. This is a cross-sectional questionnaire cohort study in people with ectodermal dysplasia. Please revise.
Line 67: ‘summasry’ should read summary
Regarding the magazine, is a magazine aimed to the support group “Ektodermale Dysplasie e.V.”? Or is open to any natural person interested in ectodermal dysplasia? Generally speaking, what is the magazine’s content?
Line 80: ‘The participants had to be at least 16 years old’. I would describe this as ‘adults participants aged 16 and above were included in this study.’
Lines 82 and the following paragraph. Data source are the participants of the study. What you are describing here are the variables and the outcome measure. Moreover, this section contains information described previously. Consider editing or delete this section.
What is the justification of choosing 50% as a minimum sample size?
Results
This section is repetitive. Many subheadings can be combined in one only section. Including a special section on statistical analysis which has been described previously is redundant.
Table 1 is a figure.
I am confused by the question ‘Satisfaction with the support provided by the health care system’. The word ‘support’ might need some explanation. Is this question related to dental care access? Satisfaction with dental treatment? Please expand.
It would be valuable if OHIP scores were presented in a table including mean, standard deviation, median and statistical significance (and the statistical test used), but also including the OHIP subscales scores. Authors are missing a great opportunity of presenting valuable information obtained from this questionnaire.
Table 2 would be more understandable if presented in a histogram. In this way, you can graphically represent distributions’ shape. Remember that you are testing for differences in distributions.
It would be interesting to describe in detail the correlation analysis in the results section and discussed afterwards. For example, what was the correlation between OHIP and satisfaction with dental services?
Figure 1
I do not agree with the term ‘externally inconspicuous’. Maybe is better referring to ‘no visible signs of ectodermal dysplasia’ instead. However, I would include patients photographs only for a very good reason, which is not the case of this manuscript. However, this as a subjective judgement.
Discussion
In my opinion, with a convenience sample size and no power calculation is not fair, referring to this response rate as a ‘high’.
It is highly likely that people with ectodermal dysplasia consulting for dental care, have been medically diagnosed with their condition during early childhood. Thus, the orthodontist playing a decisive role in the recognition of patients with ectodermal dysplasia is not a valid argument to be discussed here.
What about other aspects of peoples’ quality of life? From the OHIP questionnaire, information can be found about, for example, the psychological and social aspects of oral health that can be discussed in this section, but this manuscript refers only to clinical aspects of oral health.
Conclusion
The statement ‘People with ectodermal dysplasia are affected by a worse OHRQoL than is usually prevalent in the normal population’ is arguable. Self-assessments of QoL provide with ratings of QoL as affected by health. Thus, individuals are not affected by a worse QoL, but hey rate their QoL as better or worse than the rest of the population.
Author Response
We would like to thank the Editor and the Reviewers for revising our manuscript [ijerph-509304] entitled “Self-assessment of oral health related quality of life in people with ectodermal dysplasia“ and the constructive points discussed. The helpful comments and suggestions for improving the manuscript have been incorporated into the revised version and all changes were highlighted “Track Changes”. In this letter, we provide a point-by-point response to each addressed comment and hope the manuscript is now suitable for publication in the International Journal of Environmental Research and Public Health.
Reviewer 3:
Comments and Suggestions for Authors
This manuscript describes the oral health related quality of life of people with ectodermal dysplasia in Germany. As this journal is international in scope, I would suggest the authors add the location to the title reflecting the focus area.
We changed the title to: “Self-assessment of oral health related quality of life in people with ectodermal dysplasia in Germany“
The study provides with limited information due to the specify cohort selected and the way of collecting data.
Overall, it is written in good English.
Introduction
Lines 43-43: ‘but also forms including cleft formation including EEC syndrome’. What do you mean by ‘forms’? This sentence is confusing. Please revise.
We revised this sentences and also changed “forms” to “subtypes” (line 42):
“Also subtypes including cleft formation like EEC syndrome (Ectrodactyly-ectodermal dysplasia-cleft lip/palate syndrome) are known”
As you mention (lines 47-48) this work has been previously described within several publications of the same research group. Thus, the novelty of the analysis described in this paper is debatable.
The methodology (questionnaire) has already been used in other studies, that is correct. However, so far no such extensive data collection on oral health-related quality of life in people with ectodermal dysplasia has been carried out. The data presented are quite new and have been collected in a study based on our previous studies.
Methods
The first paragraph is redundant with the following paragraphs explaining the study design. Please revise and rephrase.
We revised this paragraph (line 55-62):
„The study design used on this study was already described on previous works [15, 16, 17]. The study was designed as a cross-sectional questionnaire cohort study in people with ectodermal dysplasia to evaluate their respective OHRQoL. A questionnaire was developed for this purpose which consisted of questions on age, sex, age at diagnosis, the time period between the first symptoms and the diagnosis of the disease, oral symptoms, difficulties in finding a dentist for treatment and satisfaction with the level of support provided by the German health care system with regard to dental care. Furthermore, the validated German version of the OHIP-14 questionnaire was used to evaluate each individual OHRQoL“
The description of the study design is incorrect. This is a cross-sectional questionnaire cohort study in people with ectodermal dysplasia. Please revise.
We revised this paragraph (line 55-57):
„The study design used on this study was already described on previous works [15, 16, 17]. The study was designed as a cross-sectional questionnaire cohort study in people with ectodermal dysplasia to evaluate their respective OHRQoL.
Line 67: ‘summasry’ should read summary
We changed this.
Regarding the magazine, is a magazine aimed to the support group “Ektodermale Dysplasie e.V.”? Or is open to any natural person interested in ectodermal dysplasia? Generally speaking, what is the magazine’s content?
The magazine is the member magazine of the self-help group. The magazine informs about activities of the self-help group and generally about activities in the field of ectodermal dyplasia (e.g. politics, medicine, health system). We added now some more information about the magazine in the methods section (line 70-72).
Line 80: ‘The participants had to be at least 16 years old’. I would describe this as ‘adults participants aged 16 and above were included in this study.’
We changed this sentence to “Adults participants aged 16 and above were included in this study.“ (line 78)
Lines 82 and the following paragraph. Data source are the participants of the study. What you are describing here are the variables and the outcome measure. Moreover, this section contains information described previously. Consider editing or delete this section.
We decided to delete this paragraph.
What is the justification of choosing 50% as a minimum sample size?
We consider 50% to be a large number of participants. This also gave us the opportunity to stop the data collection at some point. To date we have only received 2 more questionnaires, which we have therefore not included.
Results
This section is repetitive. Many subheadings can be combined in one only section. Including a special section on statistical analysis which has been described previously is redundant.
We deleted all the subheadings and added some more information on the three dimensions of the OHIP-values in the results section (line 129-131).
Table 1 is a figure.
We changed “Table to “Figure”
I am confused by the question ‘Satisfaction with the support provided by the health care system’. The word ‘support’ might need some explanation. Is this question related to dental care access? Satisfaction with dental treatment? Please expand.
This refers to whether the participants are satisfied with the services offered and financed by the health system. We have now explained this further in the manuscript (line 116-119).
It would be valuable if OHIP scores were presented in a table including mean, standard deviation, median and statistical significance (and the statistical test used), but also including the OHIP subscales scores. Authors are missing a great opportunity of presenting valuable information obtained from this questionnaire.
The suggestions were accepted and the respective information was added to the Table 1(line 160). In the present study, only the total score was evaluated, since a separate evaluation of all three dimensions is not recommended [John MT, Miglioretti DL, LeResche L, Koepsell TD, Hujoel P, Micheelis W. German short forms of the Oral Health Impact Profile. Community Dent Oral Epidemiol. 2006 Aug;34(4):277-88]. We have now listed the mean values of all three dimensions in the results section (line: 129-131)
Table 2 would be more understandable if presented in a histogram. In this way, you can graphically represent distributions’ shape. Remember that you are testing for differences in distributions.
Table 2 was replaced by figures 2 and 3 (line 136-142).
It would be interesting to describe in detail the correlation analysis in the results section and discussed afterwards.
The results were better detailed:
“Participants who reported difficulty to find a dentist, such as those unsatisfied with the health system showed higher OHIP-values, which represents a negative impact on the oral health-related quality of life. The participants gender did not influence the OHIP-values (Table 1). There was no correlation between OHIP-values and the participants age (0.063; p= .527), age at diagnosis (-0.061; p=.533), time between first symptom and diagnosis (0.105; p= .315). (line 149-153).
and discussed in the discussion section :
„Fortunately, most of the participants (72.65%) reported that they had no difficulty finding a dentist to treat them. We could already prove in another study that people with rare diseases in Germany had no problem to find a dentist [16], even if the values now determined here were somewhat worse than in the investigation at that time (79.47%). This may be due to the complex dental therapy that some patients with oligodontia require (e.g. bone augmentation, implants, prosthesis). Maybe therefore participants who reported difficulties in finding a dentist reported worse OHIP-values“ (line 211-217).
Figure 1
I do not agree with the term ‘externally inconspicuous’. Maybe is better referring to ‘no visible signs of ectodermal dysplasia’ instead. However, I would include patients photographs only for a very good reason, which is not the case of this manuscript. However, this as a subjective judgement.
We changed to “no visible signs of ectodermal dysplasia”. We have included the pictures in the manuscript, because we want to show that the affected ones are not always easy to recognize with ED
Discussion
In my opinion, with a convenience sample size and no power calculation is not fair, referring to this response rate as a ‘high’.
You are right, we deleted this.
It is highly likely that people with ectodermal dysplasia consulting for dental care, have been medically diagnosed with their condition during early childhood. Thus, the orthodontist playing a decisive role in the recognition of patients with ectodermal dysplasia is not a valid argument to be discussed here.
Unfortunately, some patients with ED are not diagnosed that easily and it is not always the case that medical doctors diagnose those patients, because there are no typical signs of ED. Therefore, in Figures 2-7 we want to show a patient who has not those typical signs of ED. Just like this patient: normal body hair, normal sweating but only oligodontia. There are lot of patients who are diagnosed only due to their oligodontia. Typically those patients are diagnosed very late and in most cases it is the orthodontist or the dentist who recognizes the oligodontia and is questioning why there is actually an oligodontia. Unfortunately, there are still lots of patients who have been in orthodontic treatment for many years because of their oligodontia and not every orthodontist is aware of this rare disease. So we think it is very important to show those pictures- just to make aware that patients with an oligodontia can be affected by ED and the doctors shall question the reason of the oligodontia in order to help those patients.
What about other aspects of peoples’ quality of life? From the OHIP questionnaire, information can be found about, for example, the psychological and social aspects of oral health that can be discussed in this section, but this manuscript refers only to clinical aspects of oral health.
In the present study, only the total score of the OHIP-14 was evaluated, since a separate evaluation of all three dimensions is not recommended [John MT, Miglioretti DL, LeResche L, Koepsell TD, Hujoel P, Micheelis W. German short forms of the Oral Health Impact Profile. Community Dent Oral Epidemiol. 2006 Aug;34(4):277-88]. We have now added the mean values of all three dimensions in the results section:
“A separate analysis of the three individual dimensions of the OHIP-14 showed a mean value for both genders in dimension 1 (functional limitations) at 3.27 points, in dimension 2 (psychological aspects) at 6.41 points and in dimension 3 (physcial aspects) at 2.27 points.” (line: 129-131).
But since a separate evaluation of the dimensions is not recommended, we have not discussed them.
Conclusion
The statement ‘People with ectodermal dysplasia are affected by a worse OHRQoL than is usually prevalent in the normal population’ is arguable. Self-assessments of QoL provide with ratings of QoL as affected by health. Thus, individuals are not affected by a worse QoL, but hey rate their QoL as better or worse than the rest of the population.
We changed this sentence to “People with ectodermal dysplasia rate their OHRQoL worse than is usually prevalent in the normal German population“. (line 229/230)
Round 2
Reviewer 3 Report
This new version of the manuscript has been greatly improved. Thank you to the authors for taking into account the suggestions.